# Design of the Floating Hologram Method with a Reverse Pyramid Type for CT and MR Diagnosis in Clinical Room

**DOI:** 10.3390/diagnostics12051157

**Published:** 2022-05-06

**Authors:** Minchan Kim, Kicheol Yoon, Kwang Gi Kim

**Affiliations:** 1Department School of Medicine, College of Medicine, National Cheng Kung University, Tainan City 704, Taiwan; kormd98@naver.com; 2Medical Devices R&D Center, Gachon University Gil Medical Center, 21, 774 Beon-gil, Namdong-daero, Namdong-gu, Incheon 21565, Korea; kcyoon98@gachon.ac.kr; 3Department of Biomedical Engineering, College of Medicine, Gachon University, 38-13, 3 Beon-gil, Dokjom-ro 3, Namdong-gu, Incheon 21565, Korea; 4Department of Biomedical Engineering, College of Health Science, Gachon University, 191 Hambak-moero, Yeonsu-gu, Incheon 21936, Korea; 5Department of Health Sciences and Technology, Gachon Advanced Institute for Health Sciences and Technology (GAIHST), Gachon University, 38-13, 3 Beon-gil, Dokjom-ro, Namdong-gu, Incheon 21565, Korea

**Keywords:** floating hologram, observed 3D tissue, PACS, total reflection glass material, pyramid type

## Abstract

In the field of medical diagnosis, big data and three-dimensional (3D) imaging diagnosis technology are being applied due to the development of these technologies. Using radiology diagnosis methods, medical staff are increasing their understanding and ability to explain symptoms to patients, but they are experiencing difficulties due to communication problems. Therefore, if the medical staff shows the lesion by providing the patient with a 3D image, the understanding of the patient can be increased. This paper proposes the design of a system to produce an inverted pyramid-shaped floating holographic image to increase the patient’s understanding. The hologram system consists of an optical source generator and a beam mirror and utilizes a technology to plot an image using a 45° refraction angle of the beam of the optical source. Selected objects for observation were liver, colon, and lung, and to observe these tissues, a Computed Tomography (CT) image was input to the hologram system through the picture archiving and communication system (PACS), and the image was displayed. Tissues observed through the mirror can be observed from the left, right, front, and back with a 360° anterior view. Therefore, it is possible to observe at the desired position by the medical staff and the patient in the treatment room, and the image is large and clear, so it is very satisfying to observe. As a holographic imaging diagnostic system, it is expected that this study can be used in clinics, medical education rooms, and operating rooms in the future.

## 1. Introduction

In the technology of the fourth medical industrial revolution, medical imaging is rapidly developing. As diagnostic systems, radiology, augmented reality/virtual reality (AR/VR), and optical diagnostic equipment are being developed. Optical diagnostic devices used in hospitals can be easily developed individually in daily life due to the development of wearable technology based on the Internet of things (IoT) [1]. In this case, IoT wearables can monitor health conditions such as diabetes, blood pressure, temperature, heart rate, and activity levels. Therefore, wearable health condition diagnosis can be a link between preventive medicine and diagnostic medicine. Technological advances in wearable devices can enhance the externally accurate monitoring visual effect through Virtual Sensitized Reality (AR/VR) and hologram technology [2]. Among them, holograms are classified into hologram methods similar to 360° real holograms, and real holograms have not yet been commercialized as a technology for visualizing objects [3]. Pseudo-holograms are divided into floating holograms and hologram screen methods [4].

There was a problem and a blur phenomenon that the initial user recognition took time and had to move very slowly to maintain the user recognition. In addition, I showed a hologram that must be viewed from the angle when the user moves by rotating the hologram in the opposite direction according to the movement of the user, but in this process, there is a strange texture, and it is not possible to see the hologram from various angles [5]. It was possible, but it was said that it did not seem to go around the actual three-dimensional (3D) content. As a result of using a half mirror with high reflectance to improve the sharpness of the hologram, it is unfortunate that the reflection was introduced to the surrounding environment in a bright environment, and the immersion of the hologram’s appreciation was reduced [6].

In the conventional exhibition method, it is forbidden to just look at the exhibits fixed in the glass decoration hall, touch them, look back, or enlarge them. However, with such a method, it is a pity that observer can only see one fixed side of the exhibit and the observer cannot see the part the observer really wants to observe. For outdoor use, a large structure that can withstand wind pressure is required, and even if the support is maintained with the large structure, the possibility of collapse or damage due to wind pressure cannot be ruled out, so it is large at the ball party. It has some difficult points to use [7].

We are working to reduce the spatial constraints that are a problem with unidirectional floating holograms currently on the market, and the existing software team is working to integrate sound output into the inverted-pyramid-type hologram device that we have manufactured. By creating a new type of speaker product that does not have an inverted pyramid type, it is easier for medical staff to use it in the medical field, there is no problem in communicating with patients, and medical staff uses pyramid-type holograms [8].

It is expected that this will increase satisfaction. Due to technical limitations, it was used only in the pharmaceutical field, but it has been applied to the arts and entertainment fields due to technological development and improvement, and the three-dimensional effect and spatial feeling peculiar to hologram technology are popular among many people [9].

The real-life content technology that we have focused on earlier requires a device for implementation and cannot stimulate multiple senses at the same time from the viewpoint of the user’s five senses experience, and some technologies are spatial. It has the disadvantage of being highly constrained. In order to overcome this, first, the reproduction of the actual content must be given a great sense of reality, and the user must lead to perceptual and psychological immersion. One of the limitations of the research is that it was not possible to receive feedback by utilizing various items through the actual exhibition. We hope that future research will lead to the development of a more complete holographic imaging system through continuous iteration [10].

In general, floating holograms are expensive to manufacture and have the disadvantage of not being universalized, but the inverted-pyramid-shaped holograms manufactured in this experiment can be purchased at a low price rather than at an expensive price. Furthermore, it is useful in various environments such as target educational environments, exhibitions, and movie screenings. Furthermore, the purpose of this study is to make it useful for medical staff to diagnose and examine patients in a hospital environment, but the pyramid form has the disadvantage that the screen is made smaller during video production. However, considering such problems, the quality is good, the image is clear and high, and it is expected that there will be no major problems for medical staff to use holograms [11].

An experiment was conducted with an inverted pyramid hologram, and if the distance to the beam protector was short, the sharpness of the image became low, and similarly, a distorted phenomenon occurred in the production of the image.

The area of use of the hologram pyramid is not yet accurate, and there is a limit to the variety of products. In addition, there are no standards for platforms and media with mutual credibility, and it is rare for previous research to analyze pyramid hologram images [12].

The floating hologram method projects a bright image onto an inclined transparent film stream, guiding the user to observe the reflected light directly. The plotting hologram method applies a method using interference fringes so that the light looks as if an object is floating in the air [13]. Therefore, the floating hologram method can be used for diagnosing lesions in clinics or observing diseases in operating rooms [14]. In the design of a hologram, a laser beam and a special substance (film) are used [15]. However, the unit price is high, and separate 3D glasses should be worn, which is inconvenient. In contrast, floating holograms are pyramid-shaped, easy to design, visually observable, and useful for exploration in medicine, science, and education [16,17].

This paper proposes a design of a hologram device in the form of a pyramid for medical diagnosis so that lesions can be observed three-dimensionally during surgery and diagnosis can be easily explained to patients in clinics [18,19]. This paper uses the inverted pyramid method and is designed in a three-dimensional manner so that the practicioner can demonstrate lesions in three-dimensional space [20].

## 2. Analysis and Design Methods

The floating hologram structure is divided into the reflection and transmission types, as shown in Figure 1. In Figure 1a, in the reflection type, a laser beam is irradiated by a laser generator, and the irradiated laser beam is incident on a beam splitter along the path of *a* [21]. The laser beam incident from *a* is split into 0° and 90° (*θ*_q_) at *c* and *b*, respectively, and transmitted by a 45° beam splitter [17]. Therefore, the transmitted wave through *c* is an object beam, which has a refraction angle of 45° (*θ*_q_) through the mirror and reaches the film through the path of *b* [22]. Furthermore, the beam traveling through the path of *b* is a reference beam [23], and this beam reaches the film and overlaps with the object beam of *b* to expose the subject [24]. Therefore, the observer can observe the subject.

In the transmission type in Figure 1b, when the laser beam is irradiated by the laser, the object beam has a 0° (*θ*_sp_) direction through the beam splitter and reaches the mirror once along the path of *a* [25,26]. The object beam transmitted along *a* reaches the second mirror along the path of *d* through a refraction of 90° in the mirror [27]. Subsequently, the beam is transmitted to the film along the path of *e* through 90° refraction in the mirror. In addition, the reference beam reaches the third mirror along path *b* through 90° refraction in the beam splitter [28,29]. The reference beam in the mirror reaches the film along the path of *c* via a refractive index of 90° [30]. Therefore, the object and reference beams overlap in the film and form a pair [31]. However, in the hologram structure, there is a Pepper’s ghost method, which is a method of recognizing an image of an object due to light incident on a plane mirror as if the object exists at a position linearly symmetric with respect to the plane mirror [32]. Therefore, when an image is viewed using a mirror surface, a different figure can be seen depending on the line of sight, and a stereoscopic image effect appears. Floating stereoscopic images based on this principle mean seeing images floating on a screen or glass due to the reflection of light. This typical method has the characteristics of a pyramid structure and an inverted pyramid structure.

As shown in Table 1, the pyramid principle is a floating three-dimensional mounting method in which images of the front, back, left, right, and slopes are created and illuminated from each glass surface [32], creating a virtual 3D image. The imaging can also appreciate 3D images from the direction of each glass. Therefore, the quality of the virtual image is affected by the resolution and brightness of the projector and screen on which the image is projected, as well as the material of the glass. At this time, the inverted pyramid has the characteristic of being able to observe three-sided images. At this time, the small hologram pyramid may use up to one display monitor that embodies the original source image, and the large hologram pyramid may use up to four monitors.

The holographic image of the treatment room should be able to see both the front and the side. In the table, 180° (1 side) is clearly visible from one side. However, the lateral view is blocked [32]. Therefore, the medical staff can see the image in the treatment room, but the patient cannot see the image. The advantage of 270° (3 sides) is that the image can be viewed from both the front and side [32]. The 360° (4 sides) image has the advantage of being suitable for use in the treatment room because it can demonstrate a 360° view of the front, side, and rear. At this time, the observation distance between the observer and the image is 50–60 cm, and the size of the observation image is 13 cm × 13 cm.

The pyramid shape places the original source video monitor at the top, and the inverted pyramid shape places the original source video monitor at the bottom [32]. Therefore, in the case of the original source image, when the same image is arranged on both sides of the half mirror, an image suitable for the position of each side surface is separately created and arranged so that different images can be seen. Images have three dimensions (3D) meaning “space”, points meaning zero dimensions, “lines” meaning one dimension, and “planes” meaning two dimensions. The spatial concept means “three-dimensional” that is visually recognized, and if there is no concept of front, back, left, right, up, and down, the point is zero-dimensional. This is called a “plane” consisting of a set of one-dimensional “points”, and here, a “line” with only the front and back concepts is one-dimensional [32].

As shown in Figure 2, two dimensions can recognize the front, back, left, and right, but there is no concept of top and bottom. On the other hand, 3D is a set of 2D “faces”, which results in the concept of space, where all the concepts of front, back, left, right, top, and bottom exist. Therefore, the hologram image shows a virtual image in the real space and finally pursues immersion in the stereoscopic image.

Since transparent glass (hereinafter referred to as a half mirror) is used, the original source image at the top or bottom of the eye is combined with the projected background image to observe the image.

After all, the observer can observe a reflection image of only the distance of the original source image projected on the pyramid glass tilted 45°, and the monitor showing the original image is hidden at the top as shown in Figure 3a. In addition, in the case of the inverted pyramid type, as shown in Figure 3b, the monitor showing the original image is hidden underneath [32].

To the observer, the monitor from which the original source image comes out does not seem to enter the viewing angle, and by showing only the reflected image of the distance reflected on the pyramid glass tilted at 45°, it is as if it is in the center of the pyramid space. The main technology is to make the image appear to appear on the screen. At this time, an effective resolution can be obtained unless the influences of the transmittance, the surrounding environment, and the illuminance are fully utilized.

The proposed pyramid-shaped hologram structure corresponds to the reflective type, which consists of an optical source generator and a beam mirror, as shown in Figure 4a, which has a box type [33,34]. From the figure, the object beam (*obj.beam*) generated in the optical generator is refracted at the *m* position at a direction angle (*θ*_1_) of 45°, as shown in Figure 4b,c. The beam is reflected through the beam mirror and reaches the subject’s position through 45° refractions [35]. Furthermore, the reference beam provided at the *m* position of the light source generator reaches the subject via a direction angle of 0° (α = 0°). Therefore, the object and reference beams can be superposed, and the lungs can be observed by the superposition effect. Owing to the superposition, 3D images can be observed from the viewpoints of 0, 90, 180, and 270° outside the observer [36].

Four glass materials, such as subject film, half mirror, etc., were used to design the structure [37]. Thus, the fabrication of the structure is shown Figure 4d. The principle of observing the subject of the object beam with respect to a refraction angle of 45° is shown in Figure 4c and Equations (1) and (2) [7]. In contrast to (*I_new_*_1_), it appears to be distorted because of the index of refraction of 45°:(1)$$I_{new1}=\left[\begin{array}{ccc} \frac{1}{cos\theta } & 0 & dtan\theta \\ 0 & 1 & 0\\ 0 & 0 & 1 \end{array}\right]$$
(2)$$I_{new2}=\left[\begin{array}{ccc} \frac{1}{sin\theta } & 0 & -\frac{d}{tan\theta }\\ 0 & 1 & 0\\ 0 & 0 & 1 \end{array}\right]$$

In the equations, *d* is the refraction distance of the beam, and *I_new_*_2_ is the refraction image. Here, *I_new_*_1_ and *I_new_*_2_ overlap each other and cause an interference phenomenon [38].

The image in the observation direction shows the planar image *I_new_*_1_ having a refractive index between *V*_1_ and *V′*_1_ and the planar image *I_new_*_2_ having a refractive index between *V*_2_ and *V′*_2_. Therefore, the refracted image observation field may change depending on the direction, and a three-dimensional image can be observed as if it is affected by the overlap [39,40].

## 3. Experimental Results

The experiment process for proposed hologram system is shown in Figure 5. From the figure, the process for experimenting with similar floating holograms constitutes a hologram observation system in the form of an inverted pyramid, connecting notebooks capable of image provision and monitoring.

An HDMI cable connects the hologram with the laptop, and Wi-Fi connection is also available. However, for security reasons, a wired connection is recommended. In addition, the notebook is connected to a Picture Archiving and Communication System (PACS) server and can provide computed tomography (CT) or magnetic resonance imaging (MRI) images of tissues (for example, lungs, liver, colon, stomach, and brain) as shown in Figure 6. Video can be shown both in Windows and Linux environments. However, for general use, the window preferences may be appropriate. The overall picture of the holographic system was taken during the hologram experiment.

The entire structure is shown in Figure 7. As shown in Figure 8, the proposed hologram consists of the reverse pyramid type with glass films. It is fabricated using the 3D printing technique and is connected to the diagnosis system (CT and MRI) which performs imaging. Subsequently, the diagnosis image is transmitted to the hologram system through a laptop as CT and MRI files.

The overall size of the hologram system is 6.15 × 26.0 × 39.0 cm^3^, and the fabricated hologram system is shown in Figure 9. The imaging files of the lungs, liver, and colon can be obtained using the collection of medical servers with PACS tools thought the department of radiology by Gachon University Gil Medical Center in Korea.

As shown in Figure 9, the images of the lungs, liver, and colon are used for a CT program. Thus, the CT images of organs can be sent to the laptop and transmitted to the hologram system.

As shown in Equation (3), the organ is observed by the overlap (*U*_(*x*,*y*)_) of the reference and object beams via the hologram system. Here, α is the amplification of the light beam, and *θ*_(*x*,*y*)_ is the phase with respect to the focal direction distance *z* [12]:(3)$$U_{\left(x,y\right)}=\sum_{z=0}^{z}\left|\alpha \left(x,y\right)\right|e^{j\theta }$$

Figure 9 shows a photograph of the hologram system fabricated using the 3D printing technique. The 3D printing material is used for a filament with black color.

Therefore, to test the reliability and feasibility of the hologram program, imaging of the organs (large intestine, liver, and lungs), as shown in Figure 10. The images were taken using an external camera simultaneously with visual observation.

From the figure, the S-colon and the appendix, ascending colon, descending colon, and transverse colon are clearly observed in the colon, and the left- and right-side lobes are clearly visible in the liver. Hologram images of colon, liver, and lung were reproduced 3 times to obtain results for reliability verification. The results obtained through three replays showed a slight difference in resolution, but mostly the same results were obtained. The lung hologram also provides clear left and right observations to assist healthcare professionals with easy visualization when training patients or with instantly observing the state of a disease in an operating room.

## 4. Discussion

Transparencies may be used as the film material in consideration of the unit price to design a floating hologram in the form of an inverted pyramid. However, when using transparencies, the subject becomes unclear. The reason is that the reference and object beams used for illuminating the subject while reflecting the external light generate double interference with the external light. This is a scattering phenomenon of the subject due to the interference effect. However, when glass is used, the scattering effect is small owing to the peculiarity of the material, and the subject can obtain sharpness. The reason is that if total-reflection glass is used, refraction is minimized so that it is possible to find a subject that is more than twice as sharp and accurate as an OHP film. In addition, although glass is vulnerable to impact, it can be stored for a long period of time, and it is judged that it is suitable for use in doctor’s offices and operating rooms.

If the inverted pyramid hologram developed in this study is used in the medical field, it will be possible to introduce a hospital system at low cost. Looking at the types and examples of floating holograms so far, we have proposed a video system for more efficient hologram effects that goes beyond the conventional static hologram method [17].

Hologram pyramids are used in various fields of everyday life because they are inexpensive and easy to manufacture. However, even though the user can see the hologram in a polygon, if the user sees the hologram outside of the front of each side of the pyramid, the hologram can be distorted or broken. Therefore, in this paper, a depth camera is added to track the user in a desktop hologram pyramid environment where the hologram can be observed larger and more clearly, the hologram is rotated according to the user’s line-of-sight angle, and the hologram image is distorted in reverse. When we proposed a technique for producing an image, compared the result of dropping it on the hologram pyramid with the result of projecting an existing hologram image, and then projecting a newly produced hologram image, the existing hologram could not provide it. It provides a viewing angle and has improved hologram corruption and distortion issues that primarily occur where the two sides of the pyramid are connected. However, it is not considered for many users [17].

In general, a holographic system consists of an optical module, two lasers, and a beam mirror. Lasers require a high input current, and optical modules are basically expensive. The hologram system has the disadvantage of being complicated in configuration and increasing in size. However, a floating hologram consists of a beam projector, a beam mirror, and a film. Therefore, the floating hologram has advantages of being a simple device, small in size, and low cost [40].

In addition, it was not possible to observe the hologram at 360 degrees. However, if the high-performance depth camera is installed on the opposite side in the same way, users are expected to be able to view the hologram through the hologram pyramid at 360 degrees without any mounting. As for the vertical movement correction, it seems that the distortion generated up and down can be solved by calculating the y-axis distortion value and correcting the image that must be viewed at the user’s position after tracking the y position of the user’s head [41,42].

In particular, the floating hologram reverse pyramid design method produces four-dimensional images of front, back, left, and right that are reflected by the glass through a display installed on the floor, regardless of the direction of each glass. A virtual three-dimensional stereoscopic image is generated by a method that reproduces the image in three dimensions so that it can be viewed [43]. Therefore, it is possible to design the video so that it floats in the air by arranging the video on the four sides of north, south, east, and west, and projecting the video by reflecting it on each of the four sides of the pyramid [44,45].

In the proposed design method, all tissues can be observed in 3D through PACS, and data are used through the PACS server of the Gachon medical center to obtain results. All patients were treated anonymously. In the case of the liver, the morphologies of the left and right leaves are clearly observed. However, there are limits to the observations of vanes and ateliers (such as hepatics and portals), the gallbladder, and bile ducts. Furthermore, in the case of the large intestine, the blood vessels of Mucosa are not observed. The lungs are clearly observed with left and right leaf morphology, but there are limits to organ observation. These limitations are to be resolved through future software design. The proposed hologram system is expected to be used in all clinical departments and is expected to be capable of primary diagnosis and visualization for patient education.

## 5. Conclusions

This study contributes to the design of a floating hologram medical imaging diagnostic system that is easy to use in a doctor’s office or an operating room. The inverted pyramid method is used so that the floating hologram image can be observed. The inverted pyramid method can efficiently adjust the central focus and size of the image, adjusting the system size. Therefore, it is possible to operate the system in a portable form in a doctor’s office or an operating room. In addition, the hologram imaging system is a convenient and accurate reading type when observing/diagnosing lesions during surgery. Moreover, when explaining 3D stereoscopic images to patients in clinic, it is expected that reliability, comprehension, communication skills, etc., will improve, and patient education will be easier and faster for medical staff. The form of the inverted pyramid is made of glass on all four sides. Therefore, it is easy to observe the image three-dimensionally in all directions. The use of holograms in medical settings has not yet become universal. However, we hope that this research will promote their use in the medical field.

## Figures and Tables

**Figure 1 diagnostics-12-01157-f001:**
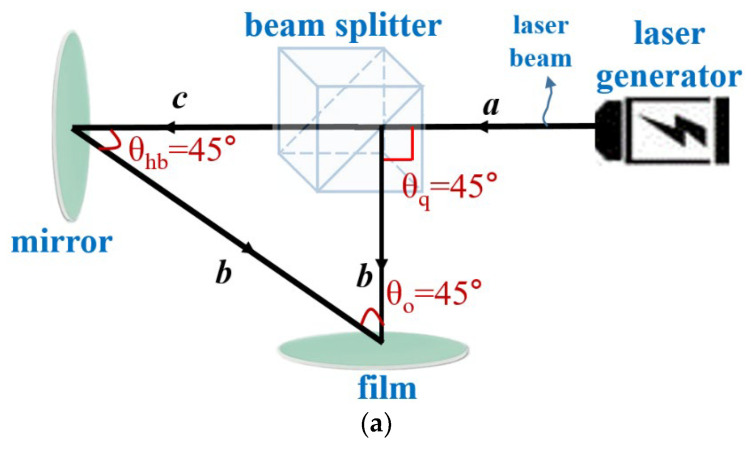
Floating hologram: (**a**) Reflective structure, (**b**) Transmissive structure.

**Figure 2 diagnostics-12-01157-f002:**
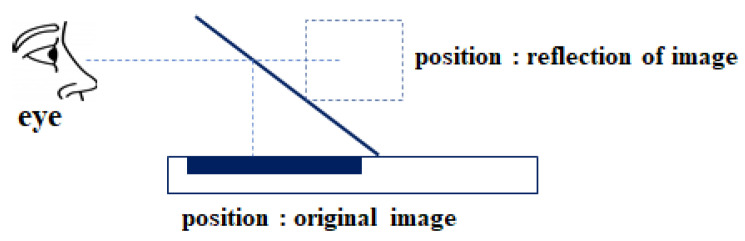
The Visible Principle of the Hologram Pyramid.

**Figure 3 diagnostics-12-01157-f003:**
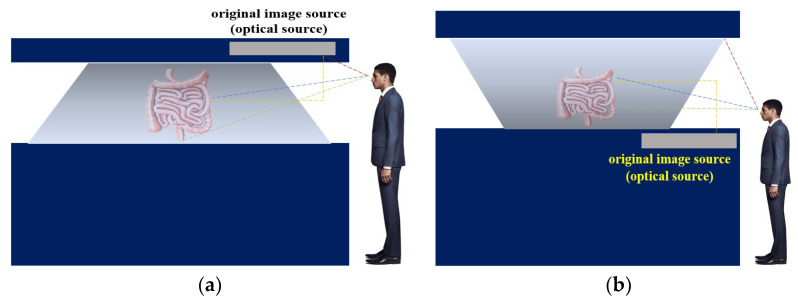
Mechanism of display arrangement using viewing angle: (**a**) Pyramid and (**b**) Inverted pyramid.

**Figure 4 diagnostics-12-01157-f004:**
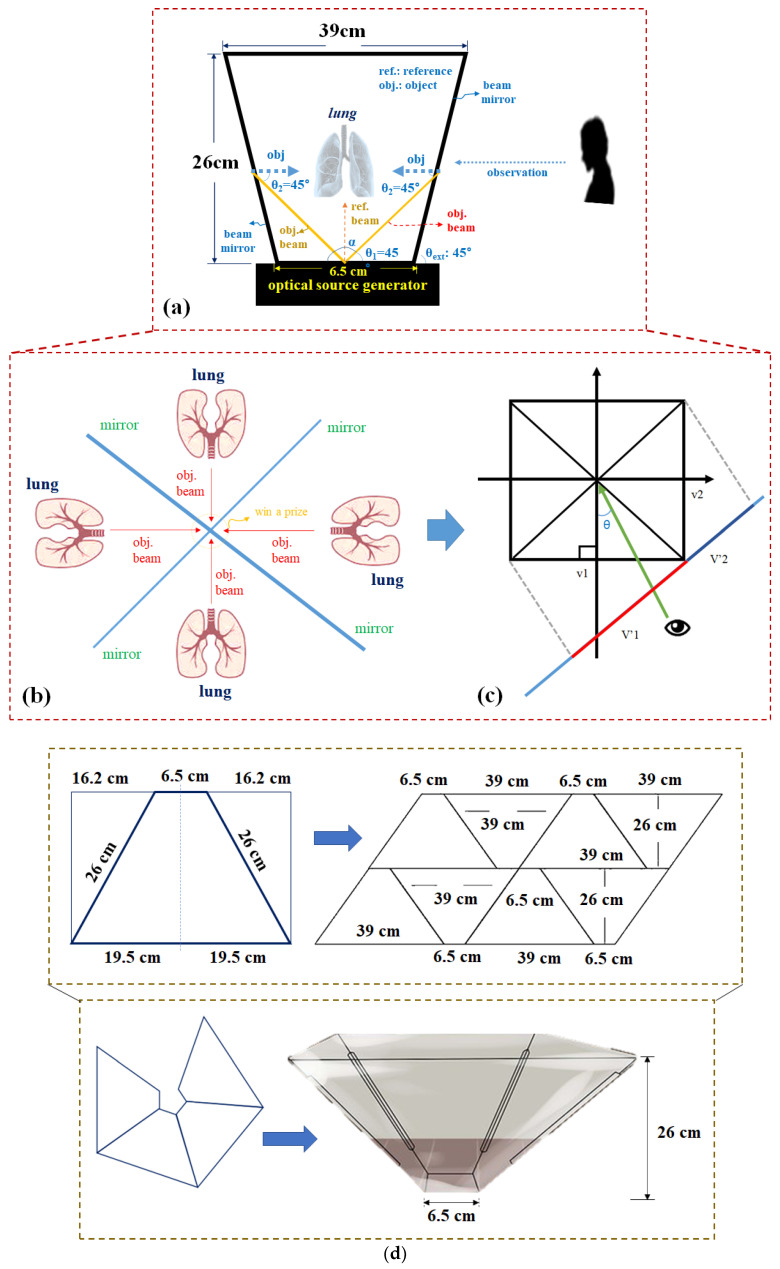
Hologram pyramid-shaped configuration and visible position (**a**) structure, (**b**) observation of four-sides (**c**) view angle of observation (**d**) process of fabrication.

**Figure 5 diagnostics-12-01157-f005:**

Flow chart of the hologram system design.

**Figure 6 diagnostics-12-01157-f006:**
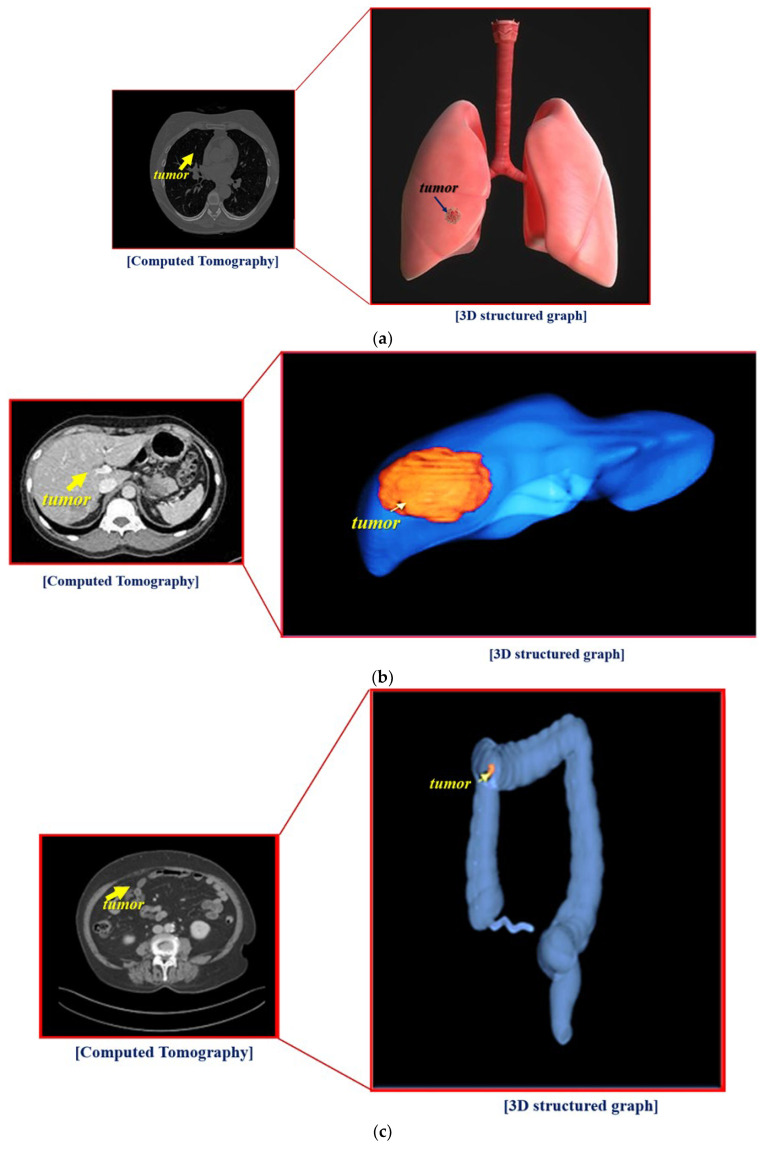
Test results obtained using the proposed hologram system: (**a**) Lung, (**b**) Liver, and (**c**) Colon.

**Figure 7 diagnostics-12-01157-f007:**
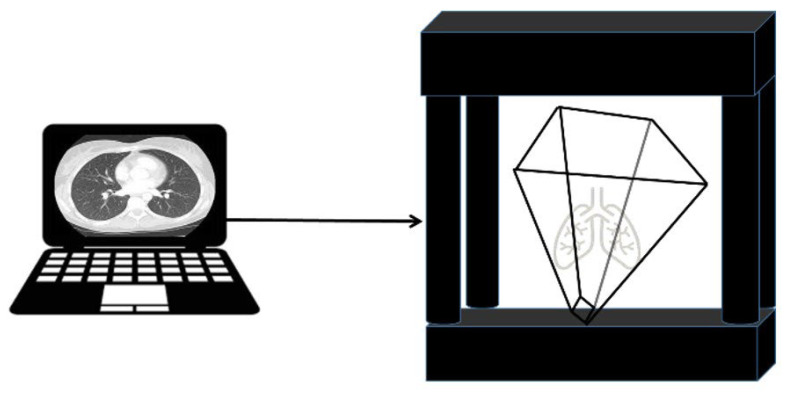
Inverted pyramid type made of an OHP film.

**Figure 8 diagnostics-12-01157-f008:**
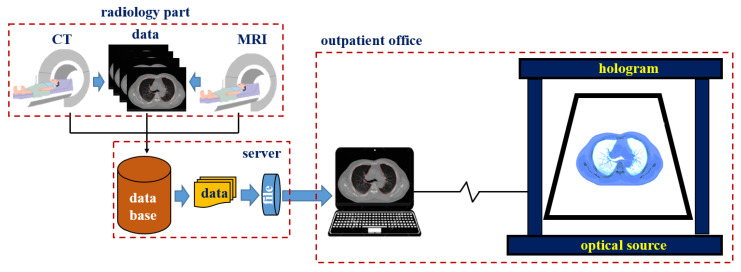
Fabricated hologram system.

**Figure 9 diagnostics-12-01157-f009:**
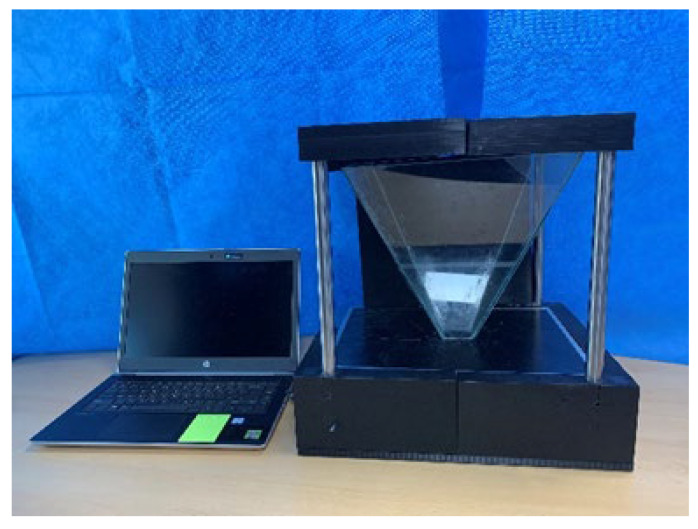
Photo of the hologram system.

**Figure 10 diagnostics-12-01157-f010:**
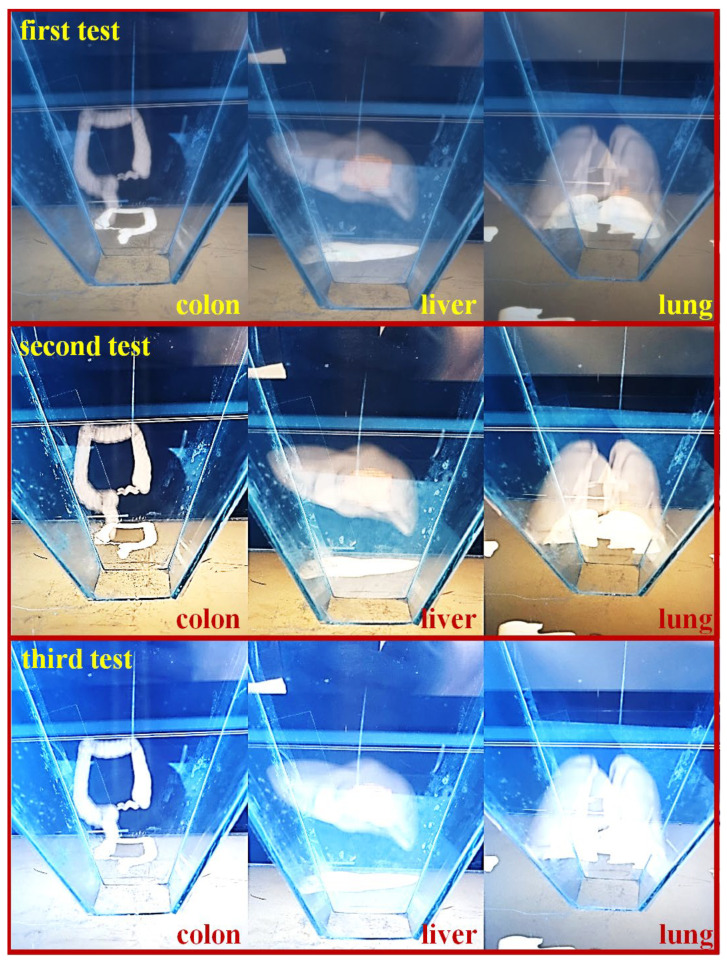
Realization of the imaging test using the proposed hologram system.

**Table 1 diagnostics-12-01157-t001:** Pyramid monitor placement and structural features, based on [32].

Monitor Placement	Division	Contents	Structure	
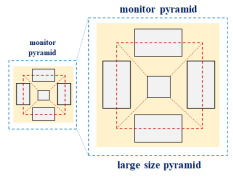	180° 1 side	One side only	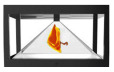	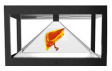
			tissue: liver	
	270° 3 sides	Three sides only	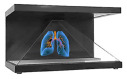	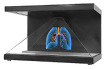
			tissue: lung	
	360° 4 sides	Four sides only	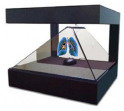	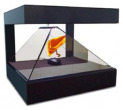
			tissue: lung	tissue: liver

## Data Availability

The data presented in this study are available upon request from the corresponding author. The data are not publicly available because of privacy and ethical re-strictions.

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
