# Peer review of "Design of the Floating Hologram Method with a Reverse Pyramid Type for CT and MR Diagnosis in Clinical Room"

_diagnostics, 2022, doi:10.3390/diagnostics12051157_

Round 1

Reviewer 1 Report

The paper is an interesting proposal to improve the diagnosis as well as the understanding of the disease or ailment for both the patient and the healthcare personnel using holograms.

I would like to make some suggestions:
- In the abstract, please define PACS.
- What happened on line 81: "General seconds"?
- All use of English in the introduction should be thoroughly reviewed by a native speaker. 
- Line 142. I think it should read "Table 1".
- In design methods, is there any limitation on the level of detail that can be shown?
- Could the pyramid and inverted pyramid methods be combined, so that the advantages of both could be taken advantage of? I am thinking of an octahedron.
- Line 216. Are you referring to Figure 5? I think the figure numbers in the text do not always correspond to the figure numbering.
- I miss having a reference of the costs of one and the other method, I would like to get an idea in terms of economic savings.
- I miss that the authors contemplate the integration of their proposal in a more ambitious environment, such as an IoT platform for control and diagnostics, for example: doi:10.3390/app804051

The paper is correct and I think it brings an interesting method. Anyway, and despite the fact that I am not a native English speaker, I found quite a few grammatical errors that hinder the reading. My main suggestion is to have it thoroughly proofread by a professional proofreader.

Author Response

Point 1: The paper is an interesting proposal to improve the diagnosis as well as the understanding of the disease or ailment for both the patient and the healthcare personnel using holograms.

I would like to make some suggestions:

Response 1: Thank you for your interest in and evaluation of our thesis. We have taken into consideration your comments.

Point 2: - In the abstract, please define PACS.

Response 2: Added to lines 28 (yellow) for your comment.

“Picture Archiving and Communication System (PACS)”.

Point 3: What happened on line 81: "General seconds"?.

Response 3: Deleted due to typo in “General seconds”. So, I made the following changes:

“Furthermore, it is”

Please refer to lines of 95 (yellow).

Point 4: All use of English in the introduction should be thoroughly reviewed by a native speaker.

Response 4: All typos have been corrected. However, English grammar is currently being corrected with the help of experts. We will try to complete the correction of English grammar before publication. Thank you for your comments.

Point 5: - Line 142. I think it should read "Table 1".

Response 5: The meaning of “Table a” is a typo. Therefore, we changed it to “Table 1”. (line : 156 yellow).

Point 6: In design methods, is there any limitation on the level of detail that can be shown?

Response 6: In design methods, is there any limitation on the level of detail that can be shown?

Point 7: Could the pyramid and inverted pyramid methods be combined, so that the advantages of both could be taken advantage of? I am thinking of an octahedron.

Response 7: Your comments are novel ideas. However, holograms for use in the medical field must have large image sizes. Because the pyramid shape projects the image from the bottom to the top, the size of the lesion becomes smaller. Therefore, the pyramid method is suitable for commercial display applications. However, since the inverted pyramid structure illuminates the image from top to bottom, the size of the image increases. Therefore, an inverted pyramid structure is suitable for holograms for use in the clinic. In addition, the reason for using an octahedron is that the 360° field of view is secured so that the patient and the medical staff can observe the image together.

Point 8: Line 216. Are you referring to Figure 5? I think the figure numbers in the text do not always correspond to the figure numbering.

Response 8 : We found that most of the figures, as well as figure 5, did not match. Thank you for taking a closer look. All corrected.

Point 9: I miss having a reference of the costs of one and the other method, I would like to get an idea in terms of economic savings.

Response 9 : The designed hologram system has the following economic benefits. And see line 329-334 (red) in the discussion.

In general, a holographic system consists of an optical module, two lasers, and a beam mirror. Lasers require high input current, and optical modules are basically expensive. The hologram system has the disadvantage of being complicated in configuration and increasing in size. However, a floating hologram consists of a beam projector, a beam mirror, and a film. Therefore, the floating hologram has advantages of a simple device, small size, and low cost.

Point 10: I miss having a reference of the costs of one and the other method, I would like to get an idea in terms of economic savings.

Response 10 : The designed hologram system has the following economic benefits. And see line 329-334 (red) in the discussion.

In general, a holographic system consists of an optical module, two lasers, and a beam mirror. Lasers require high input current, and optical modules are basically expensive. The hologram system has the disadvantage of being complicated in configuration and increasing in size. However, a floating hologram consists of a beam projector, a beam mirror, and a film. Therefore, the floating hologram has advantages of a simple device, small size, and low cost.

Point 11: I miss that the authors contemplate the integration of their proposal in a more ambitious environment, such as an IoT platform for control and diagnostics, for example: doi:10.3390/app804051.

Response 11 : Thanks for the advice. As for the diagnosis of exercise therapy using optical motion capture and IoT wearable technology, a sentence has been added to the introduction (lines 39-48: red [1]) as similarities to hologram technology are shown. Therefore, we have fully reflected your opinions. thank you.

Point 12: The paper is correct and I think it brings an interesting method. Anyway, and despite the fact that I am not a native English speaker, I found quite a few grammatical errors that hinder the reading. My main suggestion is to have it thoroughly proofread by a professional proofreader.

Response 12: All typos have been corrected. However, English grammar is currently being corrected with the help of experts. We will try to complete the correction of English grammar before publication. Thank you.

Reviewer 2 Report

The paper written by the following Authors: Minchan Kim, Kicheol Yoon, Kwang Gi Kim, entitled “Design of the floating hologram method with a reverse pyramid type for clinical diagnosis in hospital applications” presents an interesting study on a design of a hologram device in the form of a pyramid for medical diagnosis.

Although the paper is interesting, I have some major concerns:

Title

The title reflects the results presented here.

Abstract

The abstract is lacking the aim of the material and methods description as well as an informative conclusion. It should be written in more details.

Analysis and design methods

  1. There is no information about the justification for chosen objects presented in table 1. It should be included in the manuscript.
  2. What distance was between observed object and observer? It should be included in the manuscript.

Experimental results

  1. Authors analyzed lung, liver and colon. There is o information about the verification of 3d reconstructions. It should be included in the manuscript.

Author Response

Point 1: Although the paper is interesting, I have some major concerns:

Response 1: Thank you for your interest in and evaluation of our thesis. We have taken into consideration your comments.

Point 2: Title:The title reflects the results presented here.

Response 2: Changed the title as follows :

After changing the title - Design of the floating hologram method with a reverse pyramid type for CT and MR diagnosis in clinical room

Before title change - Design of the floating hologram method with a reverse pyramid type for clinical diagnosis in hospital applications.

Point 3:

Abstract:

The abstract is lacking the aim of the material and methods description as well as an informative conclusion. It should be written in more details.

Analysis and design methods

  1. There is no information about the justification for chosen objects presented in table 1. It should be included in the manuscript.
  2. What distance was between observed object and observer? It should be included in the manuscript.

Response 3:

Abstract:

Thanks for the point. The summary has been completely revised focusing on the introduction, body, results, and conclusion. Please refer to Abstract.

Analysis and design methods

  1. The selection target is added to Table 1, and justification is ref. [31] was added. I also added comments to lines 167-174 (yellow) for better understanding.
  2. The distance between the observation image and the observer is approximately 50-60 cm. And the image size is 13 cm in width and height respectively. See Lines 173-174.

Point 4 : Experimental results

Authors analyzed lung, liver and colon.

There is o information about the verification of 3d reconstructions. It should be included in the manuscript.

Response 4:

It was difficult to understand the intent of the question at first, but after a few analyzes, please check whether you understood correctly. If the answer is not enough, please explain to me again. Then we will do our best to answer you.

In consideration of your comments as much as possible, the following sentences have been constructed and the results have been added to Figure 10.

The holographic image results were taken three times for reproducibility and reliability. And I composed the following sentences (lines 292-294 / red).

“Hologram images of colon, liver, and lung were reproduced 3 times to obtain results for reliability verification. The results obtained through three replays showed a slight difference in resolution, but mostly the same results were obtained.”

Round 2

Reviewer 1 Report

Dear authors,

Thanks for your detailed reponse.

Only one indication. In "Point 11: I miss that the authors contemplate the integration of their proposal in a more ambitious environment, such as an IoT platform for control and diagnostics, for example:doi:10.3390/app804051. Response 11 : Thanks for the advice. As for the diagnosis of exercise therapy using optical motion capture and IoT wearable technology, a sentence has been added to the introduction (lines 39-48: red [1]) as similarities to hologram technology are shown. Therefore, we have fully reflected your opinions. thank you."

I added incorrectly the reference, so [1] should be:

Rodríguez-Rodríguez, I., Zamora-Izquierdo, M. Á., & Rodríguez, J. V. (2018). Towards an ICT-based platform for type 1 diabetes mellitus management. Applied Sciences, 8(4), 511.

https://doi.org/10.3390/app8040511

My fault.

Author Response

Point 1: I added incorrectly the reference, so [1] should be:

Response 1: Thank you for your kind feedback. I found the following papers and added a sentence to the introduction (lines: 38-47 / yellow).

And I added reference[1]. Thank you.

Rodríguez-Rodríguez, I., Zamora-Izquierdo, M. Á., & Rodríguez, J. V. (2018). Towards an ICT-based platform for type 1 diabetes mellitus management. Applied Sciences, 8(4), 511.

https://doi.org/10.3390/app8040511

Reviewer 2 Report

I accept the manuscript in the present form.

Author Response

Thank you for your accept decision of my paper.